# A bioinspired sequential energy transfer system constructed via supramolecular copolymerization

Yifei Han[1], Xiaolong Zhang[1], Zhiqing Ge[1], Zhao Gao[2], Rui Liao[1✉] & Feng Wang [1✉]

Sequential energy transfer is ubiquitous in natural light harvesting systems to make full use of solar energy. Although various artificial systems have been developed with the biomimetic sequential energy transfer character, most of them exhibit the overall energy transfer efficiency lower than 70% due to the disordered organization of donor/acceptor chromophores. Herein a sequential energy transfer system is constructed via supramolecular copolymerization of σ-platinated (hetero)acenes, by taking inspiration from the natural light harvesting of green photosynthetic bacteria. The absorption and emission transitions of the three designed σ-platinated (hetero)acenes range from visible to NIR region through structural variation. Structural similarity of these monomers faciliates supramolecular copolymerization in apolar media via the nucleation-elongation mechanism. The resulting supramolecular copolymers display long diffusion length of excitation energy (> 200 donor units) and high exciton migration rates (~$10^{14}$ L mol$^{-1}$ s$^{-1}$), leading to an overall sequential energy transfer efficiency of 87.4% for the ternary copolymers. The superior properties originate from the dense packing of σ-platinated (hetero)acene monomers in supramolecular copolymers, mimicking the aggregation mode of bacteriochlorophyll pigments in green photosynthetic bacteria. Overall, directional supramolecular copolymerization of donor/acceptor chromophores with high energy transfer efficiency would provide new avenues toward artificial photosynthesis applications.

[1] CAS Key Laboratory of Soft Matter Chemistry, Department of Polymer Science and Engineering, University of Science and Technology of China, Hefei, Anhui 230026, P. R. China. [2] Shaanxi Key Laboratory of Macromolecular Science and Technology, School of Chemistry and Chemical Engineering, Northwestern Polytechnical University, Xi'an 710072, P. R. China. ✉email: rliao@ustc.edu.cn; drfwang@ustc.edu.cn

Sequential energy transfer is ubiquitous in natural light-harvesting systems (LHSs) to harvest solar energy essential for living organisms[1–5]. In most photosynthetic organisms such as plants, algae, and photosynthetic bacteria, the rigid protein scaffolds serve as key elements to bind pigments and control their excitation energy transfer. Until now, a variety of artificial sequential energy transfer systems have been developed to mimic nature, by anchoring donor/acceptor (D/A) chromophores to the scaffolds including vesicles[6–8], micelles[9,10], macrocycles[11,12], biomacromolecules[13–19], and supramolecular gels[20–23]. However, most of the examples exhibit the overall energy transfer efficiency ($\Phi_{overall}$) lower than 70%[6–23], lower than the natural LHSs of purple photosynthetic bacteria ($\Phi_{overall}$: close to unity)[5]. It is primarily ascribed to the disordered D/A organization in these scaffold-supported artificial systems[24]. In this context, LHSs from green photosynthetic bacteria provide an alternative natural prototype[3], in which the bacteriochlorophyll pigments are encapsulated into the hydrophobic environment provided by chlorosomes. Direct supramolecular aggregation of light-harvesting pigments exists in chlorosomes without the aid of protein scaffolds[25–28], leading to the formation of densely packed nanostructures with the involvement of hundreds of thousands of bacteriochlorophyll (BChl) pigments (e.g., BChl C in Fig. 1a)[29–32]. Strong excitonic coupling between the individual pigments results in impressive light-harvesting capability, facilitating sequential energy transfer from BChl c antennae via BChl a in the baseplate of chlorosome to BChl a in the reaction center[3,33].

Taking inspiration from green photosynthetic bacteria, it is intriguing to aggregate D/A chromophores into long-range-ordered supramolecular copolymers with sequential energy transfer character[34–36]. The D/A chromophores can be modularly incorporated into the monomeric structure with the avoidance of tedious synthesis. Supramolecular copolymerization facilitates high D/A ratios in dynamic nanostructures. In order to achieve high sequential energy transfer efficiency in the resulting supramolecular copolymers, the elaborate choice of D/A chromophores is of crucial importance. We envisage that σ-platinated (hetero) acenes represent suitable candidates since various acene and heteroacene derivatives can be synthesized with broad spectral coverage[37–39]. These σ-platinated acenes are prone to aggregate with each other due to their structural similarity, guaranteeing the proximity of D/A pairs in the resulting supramolecular copolymers[40–42]. The Pt(II) units prevent severe stacking of (hetero)acenes to avoid aggregation-caused emission quenching, and thereby maintain sufficient emission intensity at the aggregated state.

In this work, a sequential energy transfer system with a high $\Phi_{overall}$ value has been constructed via supramolecular copolymerization of three different σ-platinated (hetero)acenes. Two amide groups are introduced to the designed monomers, which enhance non-covalent packing strength and thereby facilitate supramolecular copolymerization. In the resulting ternary supramolecular copolymers, the green-emissive species serves as a donor matrix due to its dense packing character, capable of sensitizing NIR-emissive signal via sequential D/A energy transfer (Fig. 1b). The close D/A spacing at nanometer scales and the high D/A ratio guarantee the overall sequential energy transfer efficiency of 87.4%, superior to those of the previous scaffold-supported supramolecular systems ($\Phi_{overall}$: <70%)[6–23]. Therefore, by taking green photosynthetic bacteria as the natural prototype, supramolecular copolymerization of D/A-type σ-platinated (hetero)acenes represents an effective way to promote sequential energy transfer efficiency.

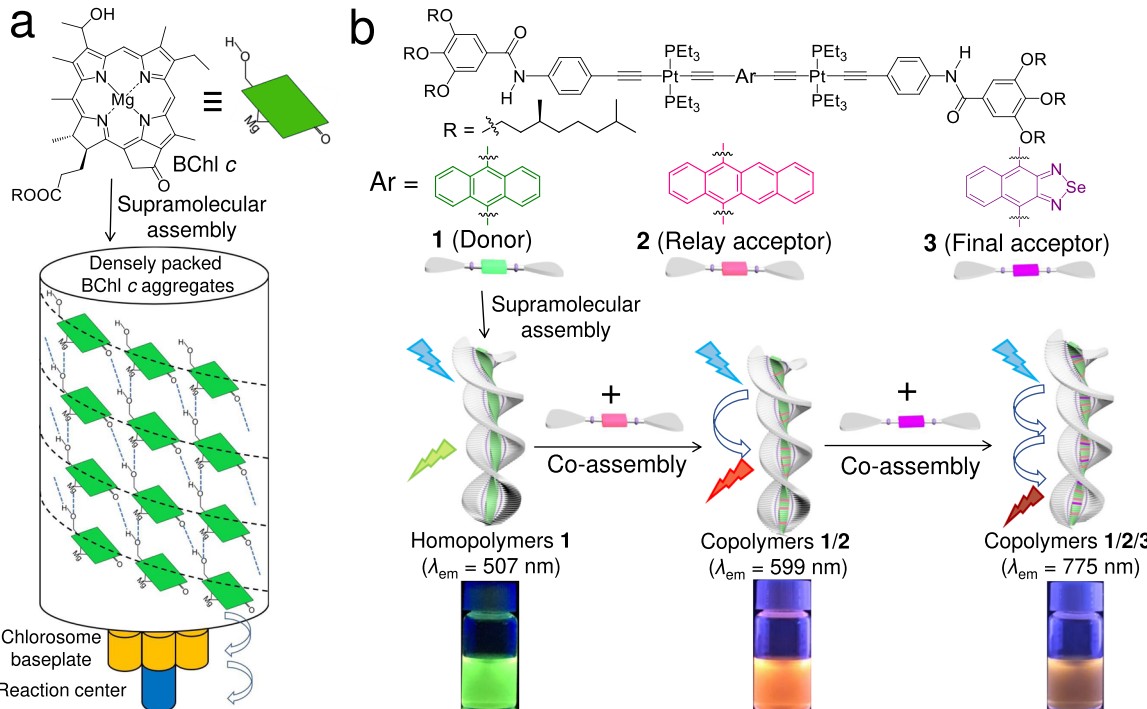

**Fig. 1 Natural and artificial sequential energy transfer systems. a** Direct supramolecular aggregation of bacteriochlorophyll c (BChl c) into a light-harvesting antenna, together with the sequential energy transfer process in green photosynthetic bacteria. **b** Supramolecular copolymerization of **1**, **2**, and **3** (cartoon symbols with green, pink, and purple colors in the middle parts, respectively) with the sequential energy transfer behaviors. The supramolecular homopolymers of **1** emit green light centered at 507 nm. The binary supramolecular copolymers **1/2** display orange emission ($\lambda_{max}$ = 599 nm) with a one-step energy transfer character. For the ternary supramolecular copolymers **1/2/3**, sequential energy transfer takes place from **1** via **2** to **3**, giving rise to the emission enhancement in the near-infrared region ($\lambda_{max}$ = 775 nm).

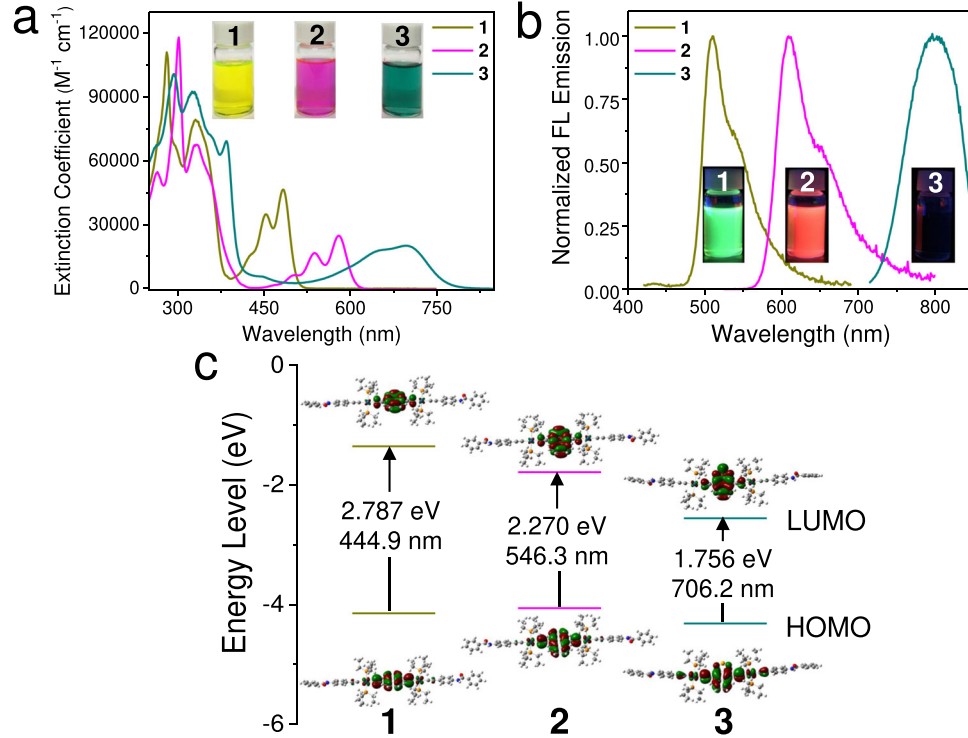

**Fig. 2 Spectroscopic characterizations of the designed compounds 1–3. a, b** Absorption and emission spectra of **1–3** (c: $1.0 \times 10^{-5}$ mol L$^{-1}$ for **1–2** in dichloromethane and for **3** in 1,2-dichloroethane), together with the corresponding photographs taken under the same concentration and solvent conditions. The inset photographs in Fig. 2b were taken under a hand-held ultraviolet lamp with an excitation wavelength of 365 nm. **c** Energy level diagram of **1–3** based on DFT computations. HOMO and LUMO represent the highest occupied molecular orbital and the lowest unoccupied molecular orbital, respectively.

## Results

**Spectroscopy of 1–3 in the monomeric state.** Specifically, monomers **1–3** (Fig. 1b) have been designed and synthesized (Supplementary Figs. 1, 2) with the incorporation of σ-platinated anthracene, tetracene, and naphtho[2,3-c][1,2,5]selenadiazole as the inner chromophores, respectively. In dilute chlorinated solvents, compounds **1–3** are dominated by the molecularly dissolved state. All of the three compounds possess strong absorbance in the ultraviolet region (see Supplementary Figs. 6–8 for the assignments of ultraviolet absorption signals), together with the moderately intense absorbance in the visible light or NIR region (Fig. 2a). In particular, for anthracene-based compound **1**, the vibronic absorption in the visible region is centered at 484 nm in CH$_2$Cl$_2$ ($\varepsilon = 4.66 \times 10^4$ M$^{-1}$ cm$^{-1}$). It bathochromic-shifts to 581 nm for compound **2** ($\varepsilon = 2.49 \times 10^4$ M$^{-1}$ cm$^{-1}$). The presence of naphtho[2,3-c][1,2,5]selenadiazole in compound **3** leads to a further bathochromic shift ($\lambda_{max} = 698$ nm, $\varepsilon = 2.02 \times 10^4$ M$^{-1}$ cm$^{-1}$). The distinct absorbances of **1–3** are directly visualized by the different solution colors like yellow-green, pink, and dark green, respectively (Fig. 2a, inset). Density functional theory (DFT) computations are employed to clarify the red-shifting absorbance from **1** to **3** (Fig. 2c). For compounds **1–2**, both HOMOs and LUMOs of electron densities are primarily distributed over the dialkynylacene units, with a minor contribution from the Pt(II) ions (Fig. 2c). Accordingly, HOMO–LUMO transitions of both compounds arise from metal-perturbed π–π* transitions of the dialkynylacene units. In terms of **3**, the electron densities of HOMO are primarily delocalized on the electron-rich naphthalene unit, which partially transfers to the electron-deficient selenadiazole unit of the LUMO orbital (Fig. 2c). Hence, the HOMO–LUMO electronic transition of **3** exhibits intramolecular charge transfer (ICT) character[43]. The calculated HOMO–LUMO energy gaps decrease from **1** (2.787 eV and 444.9 nm) via **2**

(2.270 eV and 546.3 nm) to **3** (1.756 eV and 706.2 nm), highly consistent with the experimental data (Fig. 2a). It is mainly attributed to the variation of the LUMO levels rather than the HOMO ones (Fig. 2c).

The emission signals also red-shift from **1** to **3** ($\lambda_{max}$: 510 nm of **1**, 609 nm of **2**, and 796 nm of **3**, Fig. 2b). The Stokes shift is determined to be 98 nm for **3**, larger than those of **1–2** (26 nm for **1** and 28 nm for **2**). It arises from the ICT effect of the naphtho[2,3-c][1,2,5]selenadiazole unit in **3**[43]. The quantum yield of **3** ($\Phi_F = 4.11\%$) is lower than those of **1–2** ($\Phi_F$: 48.3% for **1** and 88.4% for **2**, Supplementary Table 2), which is plausible since the NIR-emissive compound suffers from increased non-radiative deactivation rate[44]. DFT computations confirm that σ-platination of higher-order acenes render large S$_1$–T$_1$ gaps (S$_1$ and T$_1$ denote the lowest singlet excited state and the lowest triplet excited state), and thereby disfavor intersystem crossing due to the Frank-Condon barriers (Supplementary Fig. 10)[38,39]. Consequently, **1–3** exhibit fluorescent emission characters, different from the phosphorescent emission of benzene- and thiophene-substituted dinuclear Pt(II) acetylides (microsecond lifetimes) reported in the previous literature[45].

**Supramolecular homopolymerization of 1–3.** Supramolecular homopolymerization occurs for the individual σ-platinated (hetero)acene monomer in apolar media because of their strong intermolecular packing tendency. Taking monomer **2** as an example: The maximal vibronic absorption band of the dialkynyltetracene unit locates at 575 nm in methylcyclohexane (MCH, Supplementary Fig. 11b), which is 6 nm blue-shifted than that in CH$_2$Cl$_2$ ($\lambda_{max} = 581$ nm, Fig. 2a). Five isobestic points emerge upon varying the temperature (584, 551, 537, 508, and 502 nm, Supplementary Fig. 11b), revealing reversible conversion between

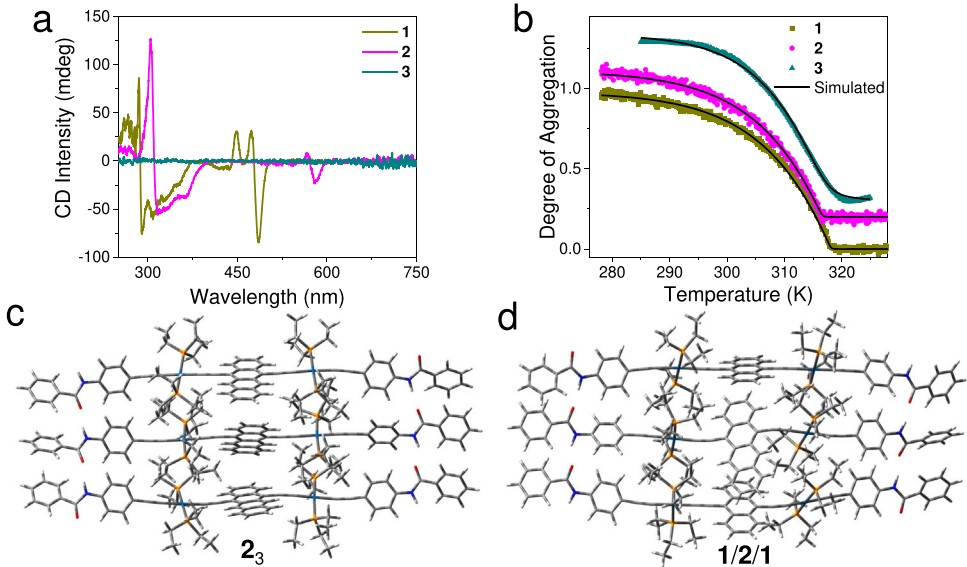

**Fig. 3 Supramolecular polymerization behaviors. a** CD spectra (1 mm cuvette, 298 K) for **1–3** in MCH (c: $2 \times 10^{-4}$ mol L⁻¹ for **1** or **2**, and $8 \times 10^{-5}$ mol L⁻¹ for **3**). The lower concentration of **3** is ascribed to its poor solubility in MCH. **b** Non-sigmoidal heating curves acquired via monitoring CD intensity changes of **1–2** (λ: 486 nm for **1** and 579 nm for **2**), together with the absorption intensity changes of **3** (λ: 709 nm) (c: $1.4 \times 10^{-4}$ mol L⁻¹ for **1** or **2**, and $8 \times 10^{-5}$ mol L⁻¹ for **3** in MCH). In panels, the curves are shown with a 0.1 offset. Data and fit are represented as colored and black lines, respectively. All melting curves were fitted by the mass balance model developed by Markvoort and ten Eikelder[47]. **c, d** Optimized geometries of trimeric species **2₃** and **1/2/1**. For both optimized geometries, non-metallic elements were described by a 6-31 G basis set, while Lanl2dz effective core potential developed by Los Alamos National Laboratory was employed to describe Pt(II) ions. Dispersion-corrected exchange functional ωb97xd was employed to optimize geometries of the trimeric species.

the monomeric and aggregated states. Depending on circular dichroism (CD) spectroscopy (Fig. 3a), a bisignate Cotton effect appears in the absorption region of tetracene units, with the positive maximum at 567 nm ($\Delta\varepsilon = 12.0$ L mol⁻¹ cm⁻¹, anisotropy factor g value = 0.00063) and the negative one at 579 nm ($\Delta\varepsilon = -35.0$ L mol⁻¹ cm⁻¹, g value = −0.00276). The result confirms the helical arrangement of σ-platinated tetracenes in MCH, as biased by the peripheral chiral (S)−3,7-dimethyloctyl chains.

The Cotton effect of **2** in MCH vanishes upon elevating the temperature to 333 K (Supplementary Fig. 13), denoting the disassembly of supramolecular polymers upon heating. By plotting $\alpha_{agg}$ (the degree of aggregation) versus temperature, a non-sigmoidal melting curve is obtained, indicating the adoption of a cooperative nucleation−elongation mechanism for the supramolecular polymerization process (Fig. 3b and Supplementary Fig. 13)[46]. The critical elongation temperature ($T_e$) is determined to be 316.5 K at the concentration of 0.14 mM (Fig. 3b). According to the mass balance model developed by Markvoort and ten Eikelder[47], the Gibbs free energy changes of elongation ($\Delta G_e$) stages is determined to be −26.3 kJ mol⁻¹ at 298 K. TEM demonstrates the formation of one-dimensional (1D) helical nanofibers with several micrometers in length (Supplementary Fig. 15b). The supramolecular stacking mode of **2** is further elucidated by DFT calculations[48]. For the optimized homo-trimer **2₃**, the monomers are held together via two-fold N−H---O hydrogen bonds between the neighboring amide units (bond lengths: 1.84–1.89 Å; bond angles: 150.4°–168.2°, Fig. 3c). It supports the crucial role of intermolecular hydrogen bonds to drive the supramolecular polymerization of **2**. Although the neighboring tetracenes adopt a face-to-face arrangement, π–π interactions are rather weak in **2₃** (π–π distances: 3.48–7.85 Å, Fig. 3c). The phenomenon is ascribed to the large steric hindrance imparted by the bulky Pt(PEt₃)₂ ligands, which prevent tight stacking of the inner tetracenes.

**Table 1 Thermodynamic parameters of the supramolecular homopolymerization of 1-3 obtained by temperature-dependent spectroscopic data[a].**

| Monomer | $\Delta H_e$ (kJ mol⁻¹) | $\Delta H_n$ (kJ mol⁻¹) | $\Delta S$ (J mol⁻¹ K⁻¹) | $\Delta G_e^b$ (kJ mol⁻¹) | $\sigma^b$ |
|---|---|---|---|---|---|
| **1** | −74.5 | −41.2 | −160 | −26.7 | $1.5 \times 10^{-6}$ |
| **2** | −74.6 | −44.1 | −162 | −26.3 | $4.6 \times 10^{-6}$ |
| **3** | −95.0 | −81.5 | −222 | −29.0 | $4.3 \times 10^{-3}$ |

[a]The thermodynamic parameters were obtained by analyzing data acquired from CD spectroscopic experiments for **1-2** and absorption spectroscopic experiments for **3**.
[b]Changes in Gibbs free energy of elongation ($\Delta G_e$) and the cooperativity factor ($\sigma$) are reported for a temperature of 298 K.

Monomer **1** or **3** is also prone to form 1D supramolecular homopolymers in MCH via the nucleation−elongation self-assembly mechanism (Fig. 3b and Supplementary Fig. 15). In terms of **1**, supramolecular chirality emerges in the supramolecular polymeric state, as reflected by the presence of the Cotton effect in the σ-platinated anthracene's absorption region (the negative maximum locates at 487 nm, Fig. 3a). The $\Delta G_e$ values are comparable between **1** and **2** (Table 1 and Supplementary Figs. 12, 13). DFT computations validate the predominant role of N−H---O hydrogen bonds to drive the supramolecular homo-polymerization processes for both compounds **1–2** (Fig. 3c and Supplementary Fig. 17a). In sharp contrast, **3** fails to induce supramolecular chirality at the self-assembled state (Fig. 3a). $\Delta G_e$ value of the **3** supramolecular homopolymerization process is larger than those of **1–2** (Table 1 and Supplementary Fig. 14). We rationalized that, in addition to intermolecular hydrogen bonds, dipole–dipole interactions[49,50] might participate in the supramolecular polymerization process. It is ascribed to the large dipole moment of the presence of the naphtho[2,3-c][1,2,5]selenadiazole heteroacene in **3** ($\mu_D$: 1.75 Debye of **3** versus 0.87 Debye for **1** and 1.20 Debye for **2**, Supplementary Fig. 16). The subtle interplay

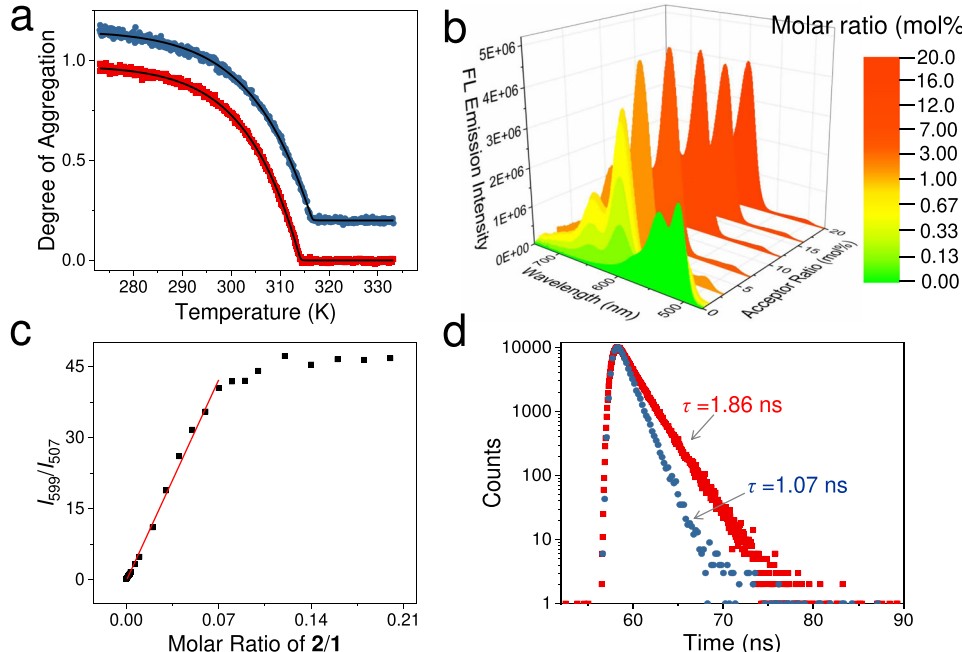

**Fig. 4 Supramolecular copolymerization and energy transfer behaviors of the binary species 1/2. a** CD melting curves of supramolecular homopolymers **1** ($c$: $8.0 \times 10^{-5}$ mol L$^{-1}$ in MCH, red line) and supramolecular copolymers **1/2** ($c$: $8.0 \times 10^{-5}$ mol L$^{-1}$ for **1** and $1.6 \times 10^{-5}$ mol L$^{-1}$ for **2** in MCH, blue line) by tracking the CD intensities at 486 nm. In panels, the curves are shown with a 0.2 offset. Data and fit are represented as colored and black lines, respectively. Both melting curves were fitted by the mass balance model developed by Markvoort and ten Eikelder[47]. **b** Steady-state fluorescent emission changes upon increasing the acceptor molar ratios. The concentration of **1** is kept at $8.0 \times 10^{-5}$ mol L$^{-1}$. Filling colors are defined based on the CIE coordinates of the emission spectra. **c** Ratiometric plot of **2** upon direct excitation of **1** ($c$: $8.0 \times 10^{-5}$ mol L$^{-1}$ in MCH). The linear fitting of the ratiometric plot is obtained by plotting the value of $I_{599}/I_{507}$ versus the molar ratio of **2**. $I_{599}$ and $I_{507}$ denote the emission intensity of **2** at 599 nm and **1** at 507 nm, respectively. **d** Fluorescence lifetime decay of supramolecular homopolymers **1** ($c$: $8.0 \times 10^{-5}$ mol L$^{-1}$ in MCH, red line) and supramolecular copolymers **1/2** ($c$: $8.0 \times 10^{-5}$ mol L$^{-1}$ for **1** and $1.6 \times 10^{-5}$ mol L$^{-1}$ for **2** in MCH, blue line).

between hydrogen bonding and dipole–dipole interactions potentially affects the non-covalent complexation mode between the neighboring monomer (Supplementary Fig. 16). As a consequence, the value of the cooperativity factor ($\sigma$) is three orders of magnitude larger for **3** than those of **1**–**2** (Table 1), indicating that the supramolecular polymerization of **3** is less cooperative.

**Supramolecular copolymerization of 1/2 with energy transfer property.** Monomers **1**–**3** are prone to stack with each other on account of their structural similarity, facilitating energy transfer for the resulting supramolecular copolymers. As a first step, supramolecular copolymerization and energy transfer behaviors are studied between **1** and **2**. With the gradual addition of **2** into the MCH solution of **1** (from 0 to 20 mol%, by keeping the concentration of **1** at $8.0 \times 10^{-5}$ mol L$^{-1}$), the CD signals of **1** at 487 nm decrease slightly for their intensities (Supplementary Fig. 18). Depending on CD melting curves, $T_e$ values elevated with the increased amounts of **2** (from 314.1 to 316.5 K, Fig. 4a and Supplementary Fig. 18). Meanwhile, the $\Delta G_e$ values are almost identical for **1** regardless of the amount of **2**, validating the heterogeneous elongation of either monomer to form randomly mixed copolymers **1/2**. DFT calculations also support the copolymerization tendency between **1** and **2**. Similar to those of the homo-trimers **1$_3$** (Supplementary Fig. 17a) and **2$_3$** (Fig. 3c), the hetero-trimer **1/2/1** is driven by N−H---O hydrogen bonds between the neighboring monomers (bond lengths: 1.82–1.87 Å; bond angles: 153.3°–168.9°, Fig. 3d). The calculated $\Delta G$ value of hetero-trimer **1/2/1** is comparable to those of the homo-trimers ($\Delta G$: −476.1 kJ mol$^{-1}$ for **1/2/1**, −473.0 kJ mol$^{-1}$ for **1$_3$**, and −466.0 kJ mol$^{-1}$ for **2$_3$**, Fig. 3c, d and Supplementary Fig. 17). The morphology of supramolecular copolymers **1/2** (Supplementary

Fig. 19) resembles that of homopolymers **1** (Supplementary Fig. 15a), both of which form entangled nanofibers depending on TEM measurements.

Since **1** features moderate emission intensity at the supramolecular polymeric state [$\Phi_F$: 19.1% for **1** ($c = 0.08$ mM) in MCH, Supplementary Fig. 20], it serves as the donor matrix to transfer energy to the encapsulated acceptor. The spectral overlap between **1** and **2** facilitates energy transfer for the resulting supramolecular copolymers at the excited state [integral: $J(\lambda) = 5.04 \times 10^{14}$ M$^{-1}$ cm$^{-1}$ nm$^4$, Supplementary Fig. 21a]. The excitation wavelength is chosen as 450 nm, in view of the weak absorption of **2** at this wavelength ($\varepsilon_{450nm}$ in MCH at 298 K: $2.72 \times 10^4$ M$^{-1}$ cm$^{-1}$ for **1** and $1.00 \times 10^3$ M$^{-1}$ cm$^{-1}$ for **2**, Supplementary Fig. 11b). With the gradual addition of **2** into **1** (from 0 to 7 mol%), the fluorescent emission band of **1** attenuates ($\lambda_{max} = 507$ nm, Fig. 4b), accompanied by the enhancement of the fluorescent emission of **2** at 599 nm (Fig. 4b). The ratiometric plot of **1/2** ($I_{599nm}/I_{507nm}$) increases linearly with the molar ratio of **2** (Fig. 4c), suggesting the dispersion of individual acceptor **2** into polymeric matrixes of **1**[23]. Upon further increasing the acceptor concentration (from 7 mol% to 20%), the $I_{599nm}/I_{507nm}$ values reach a plateau (Fig. 4c), while the 599 nm emission intensity slightly decreases for the acceptor **2** (Fig. 4b). These results support the tendency to cluster **2** in the resulting supramolecular copolymers[51]. Energy transfer between **1** and **2** is further demonstrated by the fluorescence lifetime measurements of **1** (Fig. 4d), decreasing from 1.86 ns at the supramolecular homopolymeric state to 1.07 ns at the copolymeric state (20 mol% loading of **2**). The energy transfer efficiency ($\Phi_{ET}$) is calculated to be 95.8% (Supplementary Fig. 21b), corresponding to the energy transfer rate ($k_{ET}$) of $1.23 \times 10^{10}$ s$^{-1}$ according to the Fröster mechanism.

For supramolecular copolymers **1/2**, both static and dynamic quenching participate in the fluorescent emission quenching of **1** at a high D/A molar ratio (Supplementary Fig. 23 and Supplementary Table 5). The static quenching involves the direct hetero-energy transfer between the D/A pairs, while the dynamic quenching requires homo-energy transfer within donors prior to the donor to acceptor energy transfer (Supplementary Fig. 25a). Since **1** serves as the donor matrix with densely packed character, donor-donor energy migration should be involved in supramolecular copolymers **1/2**. To determine how many donors ($n$) can be quenched by a single acceptor, a mathematical model is employed with the combination of both dynamic and static quenching mechanisms (Supplementary Equation 17)[52–55]. By non-linear plotting of the emission intensities of **1** versus the concentration of **2**, the $n$ value is calculated to be 221 (Supplementary Fig. 25b), proving the significant quenching effect rendered by the acceptor.

It is worthy to note that, in the LHSs of green photosynthetic bacteria, excitation energy delocalizes along the stacking directions of BChl aggregates (in the form of excitons) with a length up to 200 molecules[28]. Similarly, the excitation energy delocalization in supramolecular copolymers **1/2** can be illustrated via the Coulombic homo-transfer, which occurs in the form of exciton diffusion[23,56–58]. By plotting the $1/\tau$ values of **1** versus the concentration of **2** (Supplementary Fig. 25c), the second-order exciton migration rate constant of **1/2** is determined to be $6.69 \times 10^{14} \, \text{L mol}^{-1} \, \text{s}^{-1}$. It is much larger than the diffusion limit for the bimolecular reaction in solution and even higher than the exciton migration rate constant of a previously reported organic nanocrystalline-based energy transfer system ($10^{12} \, \text{L mol}^{-1} \, \text{s}^{-1}$)[59]. We rationalize that the long-range-ordered supramolecular copolymers with a high degree of polymerization and 1D structural anisotropy contribute to the accelerated exciton transport of σ-platinated acenes[60–62].

**Sequential energy transfer in supramolecular copolymers 1/2/3.** Similar to that of **1/2**, supramolecular copolymerization occurs for the binary complexes **2** and **3** in MCH (Supplementary Fig. 26 and Supplementary Table 6). Their spatial proximity induces efficient energy transfer from **2** to **3**, as reflected by the significant quenching effect ($n = 253$, Supplementary Fig. 32a) and high exciton migration rate ($9.73 \times 10^{13} \, \text{L mol}^{-1} \, \text{s}^{-1}$, Supplementary Fig. 32c). On account of the effective energy transfer properties for both **1/2** and **2/3**, we sought to investigate the sequential energy transfer behaviors for the ternary supramolecular copolymers **1/2/3**. In particular, a small amount of **3** is incorporated into supramolecular copolymers **1/2** (100: 20 mol%) as the second acceptor. Upon excitation at 450 nm, the sensitized emission of **2** centered at 599 nm attenuates (Fig. 5a), with the concomitant enhancement of NIR emission of **3** at 775 nm (Fig. 5a). The virtual Stokes shift is 325 nm for the ternary supramolecular copolymers **1/2/3**, thanks to the large gap between the maximal absorption signal of **1** and the emission signal of **3**. With the loading of **3** (10 mol%) into supramolecular copolymers **1/2**, the fluorescence lifetimes of **2** shorten from 18.4 ns to 4.49 ns (Fig. 5b). The sequential energy transfer behaviors are further verified by the two-dimensional excitation spectra difference between **1/2** (Fig. 5c) and **1/2/3** (Fig. 5d). With 450–600 nm light excitation, the emission signal of **2** at 600–725 nm undergoes substantial reduction upon ternary supramolecular copolymerization, while the resulting copolymers **1/2/3** display the NIR emission band ranging from 720 to 850 nm almost purely out of compound **3** (Fig. 5d). The phenomena suggest that efficient sequential energy transfer occurs when excited by a broad range of wavelengths from 450 to 750 nm[63]. Transient absorption measurements are further employed to characterize the sequential energy transfer in supramolecular copolymers **1/2/3**, confirming faster decays and shortened lifetimes for the transient species of both **1** and **2** (Supplementary Fig. 37).

It should be mentioned that, in addition to the two-step sequential energy transfer process, direct energy transfer from the donor to the final acceptor may also participate in the funneling of excitation energy[64]. In the ternary supramolecular copolymers **1/2/3**, we rationalized that the direct energy transfer from **1** to **3** plays a minor role in the whole energy transfer process. It is ascribed to the less D/A spectral overlapping of **1/3** than that of **1/2** ($1.90 \times 10^{14} \, \text{M}^{-1} \, \text{cm}^{-1} \, \text{nm}^4$ versus $5.04 \times 10^{14} \, \text{M}^{-1} \, \text{cm}^{-1} \, \text{nm}^4$, respectively, Supplementary Figs. 21a and 38a). Since the excitation energy is preferentially directed to the acceptor with the larger spectral overlap[1,65], the majority of excitation energy of donor **1** should flow to **2** rather than **3**. To prove this assertion, we calculate the $k_{\text{ET}}$ rates for each energy transfer step[64,66,67]. Considering that both **2** and **3** receive excitation energy from **1**, the sum of $\Phi_{\text{ET}}$ for **1/2** and **1/3** are equal to the overall ET efficiency of **1** (92.4%). In combination with the calculated $k_{\text{ET}}$ rates of **1/2** and **1/3** (**1/2**: $4.69 \times 10^9 \, \text{s}^{-1}$, **1/3**: $1.85 \times 10^9 \, \text{s}^{-1}$, Supplementary Equations 18, 19), it can conclude that the contribution of **1/3** to $\Phi_{\text{ET}}$ of **1** is 26.1%, much lower than that of **1/2** (66.3%). The detailed sequential energy transfer process of supramolecular copolymers **1/2/3** is further described by the energy level diagram (Fig. 6a). As can be seen, the $k_{\text{ET}}$ values for both **1/2** and **1/3** in the ternary supramolecular copolymers **1/2/3** decrease compared to the corresponding binary supramolecular copolymers (**1/2**: $4.69 \times 10^9 \, \text{s}^{-1}$ versus $1.23 \times 10^{10} \, \text{s}^{-1}$; **1/3**: $1.85 \times 10^9 \, \text{s}^{-1}$ versus $4.84 \times 10^9 \, \text{s}^{-1}$). Meanwhile, $k_{\text{ET}}$ accelerates between **2** and **3** ($6.69 \times 10^8 \, \text{s}^{-1}$ in the ternary supramolecular copolymers **1/2/3** versus $2.04 \times 10^8 \, \text{s}^{-1}$ in the binary supramolecular copolymers **2/3**). Remarkably, the $\Phi_{\text{overall}}$ from **1** to **3** is determined to be 87.4% ($\Phi_{\text{overall}} = \Phi_{\text{ET}1/2} \times \Phi_{\text{ET}2/3} + \Phi_{\text{ET}1/3}$), which is superior to the previously reported supramolecular sequential energy transfer system ($\Phi_{\text{overall}}$: <70%)[6–23].

It is worthy to note that supramolecular copolymerization is a prerequisite for efficient energy transfer. When **1−3** are mixed in $CH_2Cl_2$ (dominated by the monomeric state), the energy transfer efficiencies for **1** and **2** are significantly lower than those in MCH (dominated by the supramolecular copolymeric state) (**1**: 37.1 versus 92.4%, **2**: 16.7 versus 92.5%, Supplementary Fig. 36). A control experiment is further performed to validate the importance of intermolecular hydrogen bonds to direct efficient energy transfer, by employing the iodide σ-platinated (hetero)acenes **5–6** (Fig. 6b) instead of **2–3** as the acceptor units. When **5–6** are incorporated into the donor matrix of **1**, they fail to copolymerize with **1** due to the absence of amide units in their structures. Alternatively, they could potentially be encapsulated in the rigid domains of supramolecular polymers derived from **1**. As a consequence, the $\Phi_{\text{ET}}$ of **1** is 38.0% in the resulting complexes **1/5/6** (Fig. 6b), drastically lower than that of supramolecular copolymers **1/2/3** ($\Phi_{\text{ET}}$: 92.4% for **1**). $\Phi_{\text{ET}}$ of the relay acceptor also displays a dramatic reduction (14.0% for **5** versus 92.5% for **2**, Fig. 6b). Overall, it unambiguously supports the importance of supramolecular copolymerization to organize D/A-type σ-platinated (hetero)acene chromophores and thereby promote the energy migration/transfer[59].

**Photo-triggered modulation of sequential energy transfer.** As previously reported, acenes and σ-platinated acenes undergo photo-induced endoperoxidation in the presence of oxygen (Supplementary Figs. 39−40, 43)[40,41]. In this regard, we sought to modulate the sequential energy transfer process of **1/2/3** via photo-irradiation. Upon 460 nm light irradiation for 36.5 min, the CIE (Commission Internationale de l'Eclairage) coordinates

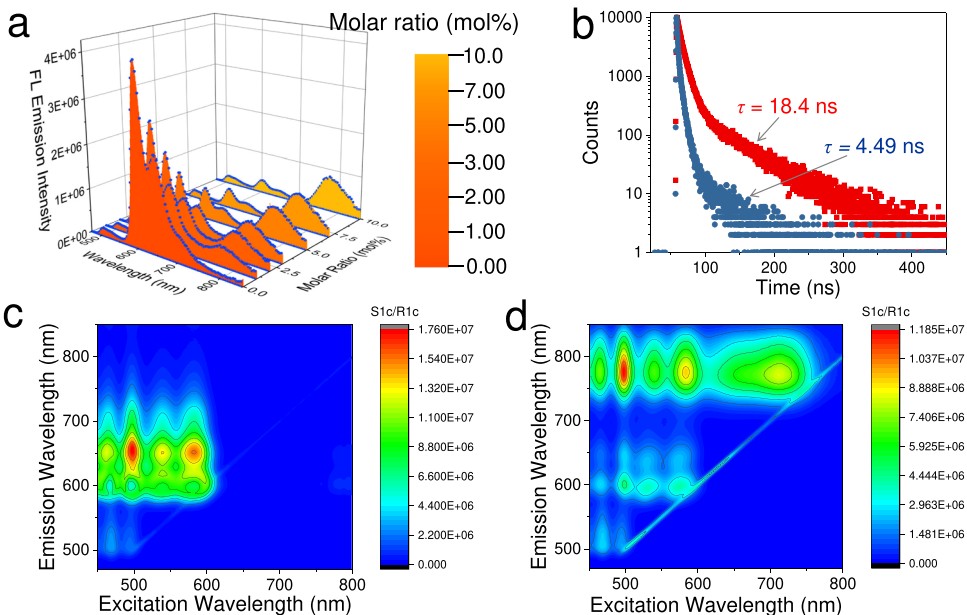

**Fig. 5 Sequential energy transfer behaviors of the ternary supramolecular copolymers 1/2/3. a** Steady-state fluorescence emission changes upon titrating **3** into supramolecular copolymers **1/2** (100: 20 mol%) (*c*: $8.0 \times 10^{-5}$ mol L$^{-1}$ for **1** and $1.6 \times 10^{-5}$ mol L$^{-1}$ for **2** in MCH). The filling colors of emission profiles are defined according to the Commission Internationale de l'Eclairage (CIE) coordinates. **b** Fluorescence lifetime decay curves of supramolecular copolymers **1/2** (100: 20 mol%) and **1/2/3** (100: 20: 10 mol%) at 599 nm (*c*: $8.0 \times 10^{-5}$ mol L$^{-1}$ for **1**, $1.6 \times 10^{-5}$ mol L$^{-1}$ for **2**, and $8.0 \times 10^{-6}$ mol L$^{-1}$ for **3** in MCH). **c** Two-dimensional excitation spectrum of the binary supramolecular copolymers **1/2**. **d** Two-dimensional excitation spectrum of the ternary supramolecular copolymers **1/2/3**.

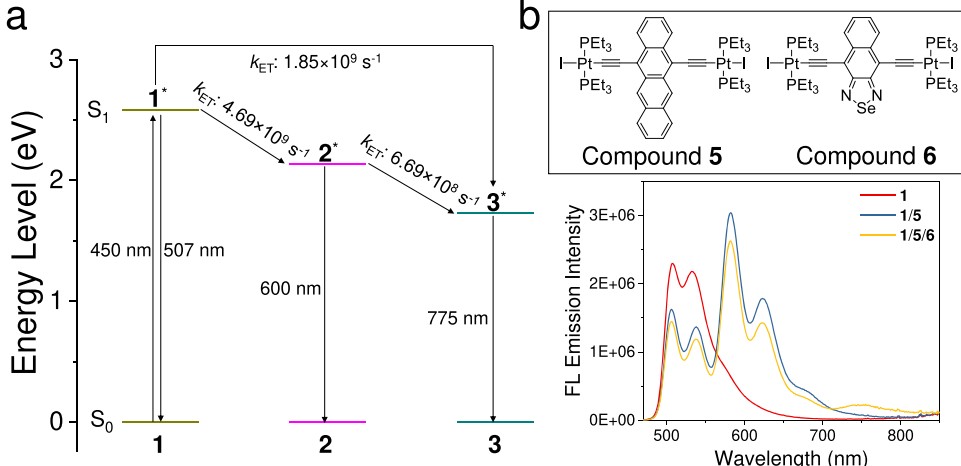

**Fig. 6 Sequential energy transfer parameters and control experiment. a** Energy level diagram for energy transfer in supramolecular copolymers **1/2/3**. It is assumed that $S_0 \rightarrow S_1$ electronic transition exclusively takes place for **1** upon 450 nm excitation. **b** Steady-state fluorescence emission changes upon titrating **5** and **5/6** into the supramolecular polymers of **1** (*c*: $8.0 \times 10^{-5}$ mol L$^{-1}$ for **1** in MCH, $1.6 \times 10^{-5}$ mol L$^{-1}$ for **5**, and $8.0 \times 10^{-6}$ mol L$^{-1}$ for **6** in MCH). The k$_{ET}$ refers to the energy transfer rate between donor and acceptor.

change from pale brown (0.491 and 0.469) to green (0.247 and 0.594). Simultaneously, the fluorescent emission signals of **2** and **3** almost disappeared, accompanied by a 25.2-fold enhancement of donor emission (Fig. 7a). The absorption bands of **2** and **3** also vanish (Supplementary Fig. 41a), confirming the faster photo-reaction rates of σ-platinated tetracene and naphtho[2,3-*c*][1,2,5] selenadiazole than that of σ-platinated anthracene on the measured timescale. The result is particularly intriguing since the individual compound **3** shows negligible self-sensitized reactions under 460 nm light irradiation (Supplementary Fig. 39c). It suggests that supramolecular copolymerization facilitates the acceleration of the photo-chemical reactions. When switching the light irradiation wavelength to 525 nm, both the absorption and

emission signals of **2** vanished (Supplementary Fig. 42a, b), while the absorption signals of **1** and **3** remain almost unchanged (Supplementary Fig. 42a). It supports the selective removal of **2** in the ternary supramolecular copolymers upon 525 nm light irradiation. Accordingly, the elaborate manipulation of the irradiation wavelengths from 460 nm to 525 nm enables the disruption of sequential energy transfer in a selective manner (Supplementary Fig. 43).

Notably, the ternary supramolecular copolymers **1/2/3** exhibit time-dependent CIE coordinate changes upon light irradiation. In detail, negligible variation of the emission color occurs upon 460 nm light irradiation for 270 s (Fig. 7b). When the irradiation time varies from 270 to 1040 s, the CIE coordinates change linearly

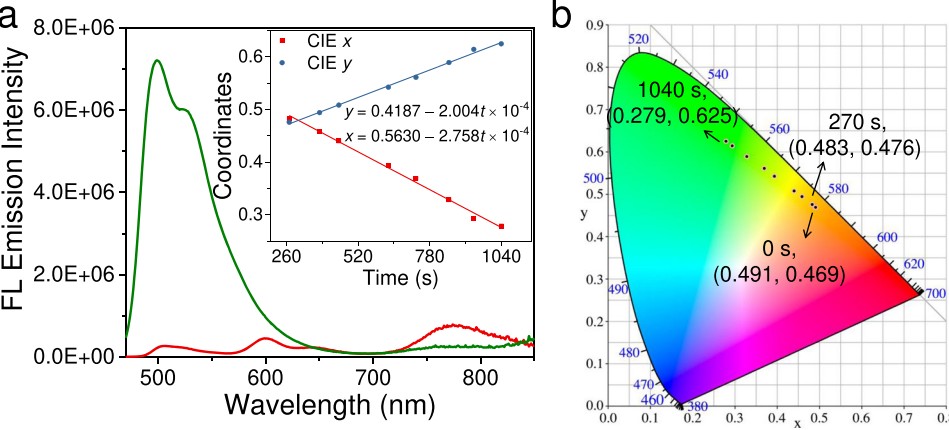

**Fig. 7 Emission color tuning of the ternary supramolecular copolymers 1/2/3. a** Fluorescent emission spectral changes of supramolecular copolymers **1/2/3** without light irradiation (red line) and with 460 nm light irradiation for 36.5 min (green line). Inset: changes of CIE $(x, y)$ coordinates versus the 460 nm irradiation time $(t)$. **b** CIE coordinate changes upon 460 nm irradiation. The measured concentrations are $8.0 \times 10^{-5}$ mol $L^{-1}$ for **1**, $1.6 \times 10^{-5}$ mol $L^{-1}$ for **2**, and $8.0 \times 10^{-6}$ mol $L^{-1}$ for **3** in MCH.

from pale brown (0.483 and 0.476) to green (0.279 and 0.625) (Fig. 7b). By plotting CIE coordinates $(x, y)$ versus irradiation time, linear relationships are established to describe CIE $(x, y)$ variations during this period, namely CIE $(x = 0.5630 - 2.758t \times 10^{-4}, y = 0.4187 + 2.004t \times 10^{-4}$, Fig. 7a, inset). Upon switching the irradiation wavelength to 525 nm, linear CIE coordinate changes also exist [CIE $(x = 0.5479 - 3.277t \times 10^{-5}, y = 0.4286 + 2.487t \times 10^{-5}$, Supplementary Fig. 42d]. For both 460 and 525 nm photoirradiation, the intercepts of $x$ and $y$ are nearly identical for the functional expressions of CIE, yet the variation rates decrease 8.42-fold and 8.06-fold for the latter wavelength, illustrating the slower CIE coordinate changes under 525 nm light irradiation. Overall, light-triggered modulation of energy transfer in supramolecular copolymers would be promising for programmable emission color tuning.

## Discussion

In summary, an artificial sequential energy transfer system has been successfully constructed via supramolecular copolymerization of σ-platinated (hetero)acenes, by taking inspiration from the natural light-harvesting systems of green photosynthetic bacteria. Through structural variation of (hetero)acenes, the transitions of vibronic absorption and emission bands can be tuned from visible to NIR region. The designed monomers **1–3** tend to assemble into one-dimensional supramolecular polymers in apolar MCH via the nucleation-elongation mechanism. Multi-component supramolecular copolymerization occurs for **1–3** due to their structural similarity, giving rise to efficient energy transfer behaviors. The binary supramolecular copolymers **1/2** and **2/3** display long diffusion length of excitation energy (>200 donor units) and high exciton migration rates ($\sim 10^{14}$ L mol$^{-1}$ s$^{-1}$). For the ternary supramolecular copolymers **1/2/3**, the sequential energy transfer takes place with a $\Phi_{overall}$ value of 87.4%, which is significantly higher than those of the previously reported supramolecular sequential energy transfer system ($\Phi_{overall}$: <70%). The superior energy transfer properties originate from the densely packed σ-platinated (hetero)acene chromophores in long-range-ordered supramolecular copolymers, mimicking the supramolecular aggregation mode of BChl pigments in green photosynthetic bacteria. Overall, directional supramolecular copolymerization of donor/acceptor chromophores with sequential energy transfer character would benefit the further development of highly efficient artificial photosynthesis systems.

## Methods

**Measurements.** All NMR spectra were measured and recorded on a Varian Unity INOVA-400 spectrometer. Tetramethylsilane (TMS) were utilized as the internal standard for [1]H NMR and [13]C NMR spectra, whilst 85% H₃PO₄ was used for [31]P NMR spectra. The MALDI-TOF spectra were measured on a Bruker Autoflex Speed spectrometer with trans-2-[3-(4-*tert*-Butylphenyl)-2-methyl-2-propenylidene]malononitrile (DCTB) as the basic matrix. UV–Vis-NIR absorption were performed on a UV–1800 Shimadzu spectrometer. FluoroMax-4 spectrofluorometer were employed to study the steady-state fluorescence emission and the two-dimensional excitation spectra. Fluorescence lifetimes were measured on a Fluorolog-3-Tau and deltaflex apparatus. Coumarin 153 ($\Phi_F =$ 0.544 in ethanol) and rhodamine B ($\Phi_F = 0.710$ in ethanol) were exploited as standard references for the quantum yields of **1** and **2**. Considering that the emission signal of **3** locates in the NIR region, cryptocyanine ($\Phi_F = 0.012$ in ethanol) was used as the standard reference for the determination of quantum yield. Full CD spectra and variable temperature CD measurements were carried out on a Jasco J-1500 circular dichroism spectrometer. The temperature was monitored and regulated by a PFD-425S/15 Peltier-type temperature controller during the CD experiments. Matlab software was employed to perform the nonlinear fitting of non-sigmoidal curves based on the mass balance model developed by Markvoort and ten Eikelder. TEM images were performed on a Tecnai G2 Spirit BioTWIN electron microscope (acceleration voltage: 100 kV). Nanosecond transient absorption spectra were performed and recorded on a helios femtosecond pump-probe transient absorption spectrometer. During the measurements, the excitation wavelength and the power of the pulse laser was 430 nm and 0.03 mW, respectively. The analysis of nanosecond transient absorption spectra was carried out by utilizing Surface Xplorer software developed for ultrafast spectroscopy analysis. To investigate the photo-triggered modulation of sequential energy transfer, LED lamps (power: 12 W) with different wavelengths (460 and 525 nm) were employed to perform the light irradiation experiments. The distance between LED lamps and irradiated samples were kept at 10 cm. To ensure the endoperoxidation of acenes, all irradiated samples were saturated with air prior to the irradiation experiments.

**Theoretical calculations.** Gaussian 09 software was utilized to perform the theoretical computations. B3LYP exchange functional was chosen to calculate the optimized geometries of **1–3**. Meanwhile, non-metallic elements (C, H, O, N, Se, and P) were described by a 6-31 G(d) basis set, while Lanl2dz effective core potential developed by Los Alamos National Laboratory was employed to describe Pt(II) ions. To study electronic transitions of compounds **1–3**, TDDFT calculations were carried out at the identical computational level without adding a solvation model. Besides, to gain deeper insights into the assembly driving forces, dispersion-corrected exchange functional ωb97xd was exploited to optimize geometries of trimeric species. To decrease the costs of the optimized trimers, the basis set for all non-metallic elements was decreased to a 6-31 G computational level. Frequency calculations demonstrated that imagery frequencies were absent for all of the optimized geometries.

## Data availability

The data that support the findings of this study are available from the corresponding author upon request.

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

## Acknowledgements

This work was supported by the National Natural Science Foundation of China (21922110 and 21871245), the Fundamental Research Funds for the Central Universities (WK3450000005), and the Starry Night Science Fund at Shanghai Institute for Advanced Study, Zhejiang University (SNZJU-SIAS-006). We are grateful for the technical support from the High-Performance Computing Center of the University of Science and Technology of China for DFT and TDDFT computations.

## Author contributions

Y.H. and F.W. conceived the idea for this manuscript. Y.H. synthesized the samples, performed spectroscopic experiments, and carried out DFT computations. X.Z., Z.Ge, and Z.Gao contributed to the emission lifetime and transmission electron microscopy measurements. R.L. contributed to the supramolecular polymerization mechanism studies. The manuscript was written and revised by Y.H., R.L., and F.W.

## Competing interests

The authors declare no competing interests.
