## [Peer Review File · Nature Communications]

REVIEWER COMMENTS

Reviewer #1 (Remarks to the Author):

In this manuscript, Wang and coworkers reported construction of an elegant light harvesting system with sequential energy transfer character via supramolecular copolymerization strategy. It is widely known that in green photosynthetic bacteria direct pigment-pigment interactions of bacteriochlorophyll units exist without the aid of protein scaffolds, which form densely packed nanostructures with high energy transfer efficiency. Inspired by the natural system, three structurally similar compounds have been designed and synthesized, with the incorporation of σ -platinated anthracene, tetracene, and naphthoselenadiazole as inner cores. The supramolecular copolymerization is proceeded via the cooperative nucleation-elongation mechanism, as driven by the combination of hydrogen bonds and π - π stacking interactions. The close donor/acceptor spacings at nanometer scale ensure long excitation diffusion lengths and high exciton migration rates. It results in a higher energy transfer efficiency even compared to that of green photosynthetic bacteria. The study presented in this manuscript shows elegantly how the energy transfer efficiency can be optimized via bio-mimetic design of the supramolecular assembled nanostructures. The work has a very good technical quality too. The spectroscopic experiments, together with DFT calculation were performed to provide the definitive evidence of the proposed results. This referee recommend this manuscript for publication in Nature Communications after considering the following comments.

1. The authors are suggested to compare the sequential energy transfer efficiency between the “good” solvent such as chloroform and the “bad” one such as methylcyclohexane. Strong differences are expected to support the importance of supramolecular copolymerization on the light harvesting behaviors.
2. In the last part: photo-responsiveness of each compound toward 460 nm and 525 nm are suggested be studied in more detail to give more information.
3. Some grammar and spelling errors are found in the main text and supplementary information. For example, “diople-dipole interactions” should be “dipole-dipole interactions”. The manuscript needs to be polished carefully.

Reviewer #2 (Remarks to the Author):

The authors describe beautiful results in a very interesting area of current science. They have used their detailed knowledge on supramolecular polymerization of Platinum-based monomers with the

concept of energy transfer. Controlled energy transfer is of great importance for all kind of concepts. I like to strongly recommend publication, but a few points need to be improved before the manuscript can be accepted for publication.

1) The term Light Harvesting System is a bit misleading for this reviewer. It is for me a cascade energy transfer. Although it is with a very high yield – worth publishing in Nat. Comm. – but it does not harvest the light in some kind of reaction. Maybe, it is well accepted, but the general audience will probably expect more. The same type of nomenclature is used in the introduction. I would suggest the authors to make clearer what are the natural systems and what are mimics and where do they differ.

2) The color coding in the figures is changing and makes following the manuscript difficult. See e.g. Figure 2a,b and Figure 3a,b.

3) The authors are using the model of Van der Schoot, many other are using today Mass Balance Models as developed by Markvoort and ten Eikelder (J. Phys. Chem. B. 2012, 116, 5291-5301).

4) Compound 3 is different from 1 and 2. Not only doesn't it show CD, also the cooperativity is much less. It comes close to an isodesmic pathway.

5) Maybe the authors can say something about the thermodynamic parameters of the copolymerization – especially when 3 is mixed with 2 and/or 1 and 2.

Reviewer #3 (Remarks to the Author):

In the current paper, the authors reported a novel artificial supramolecular SET-LHSs inspired, constructed via the supramolecular copolymerization of σ -platinated acenes. By comparing the experimental measurement with DFT and TDDFT calculation results, the authors showed that through minor structural variations of acenes, the transitions of vibronic absorption and emission bands could be readily tuned from the visible to NIR region. This result sounds reliable. However, the reviewer pays attention that in the "Result, Spectroscopy of 1–3 in monomeric state" section, the maximum peak in both of the Absorption spectra (Fig2 a) and Electronic transition spectrum (Supplementary Fig 1~3) are not located at the HOMO-LUMO transition region. These data show in the plot but lack discussion in the main text. What the excited state (S_2 , S_3 ?) and transition (HOMO-1 \rightarrow LUMO+1) are related to these peaks? Are those peaks also captured in the experimental signals? Although these peaks appear at higher energy levels, they are still within the high Violet to low UV wavelengths region and may affect the reported spectral red-shifts phenomenon. I would suggest the authors provide some discussion about those peaks. Other than that, the paper is in good shape and suitable for publishing in Nature Communications.

Feng Wang, Ph. D.

Professor,

Department of Polymer Science and Engineering

CAS Key Laboratory of Soft Matter Chemistry

University of Science and Technology of China, Hefei, 230026, P. R. China

TEL 86-551-3606095; FAX 86-551-3606095

E-mail: drfwang@ustc.edu.cn

Mar. 31, 2022

Thank you and all of the reviewers for their insightful comments on this paper (NCOMMS-21-45163)! We have made the following corrections in response to the reviewers' comments.

- 1. Reply to the first comment made by Reviewer 1 “In this manuscript, Wang and coworkers reported construction of an elegant light harvesting system with sequential energy transfer character via supramolecular copolymerization strategy. It is widely known that in green photosynthetic bacteria direct pigment-pigment interactions of bacteriochlorophyll units exist without the aid of protein scaffolds, which form densely packed nanostructures with high energy transfer efficiency. Inspired by the natural system, three structurally similar compounds have been designed and synthesized, with the incorporation of σ -platinated anthracene, tetracene, and naphthoselenadiazole as inner cores. The supramolecular copolymerization is proceeded via the cooperative nucleation-elongation mechanism, as driven by the combination of hydrogen bonds and π - π stacking interactions. The close donor/acceptor spacings at nanometer scale ensure long excitation diffusion lengths and high exciton migration rates. It results in a higher energy transfer efficiency even compared to that of green photosynthetic bacteria. The study presented in this manuscript shows elegantly how the energy transfer efficiency can be optimized via bio-mimetic design of the supramolecular assembled nanostructures. The work has a very good technical quality too. The spectroscopic experiments, together with DFT calculation were performed to provide the definitive evidence of the proposed results. This referee recommends this manuscript for publication in Nature Communications after considering the following comments. 1. The authors are suggested to compare the sequential energy transfer efficiency of the ternary species 1/2/3 in CH₂Cl₂ and MCH. Strong differences are expected to support the importance of supramolecular copolymerization on the light harvesting behaviors.”*

Thank the reviewer 1 for the kind comment on this manuscript. According to the reviewer's suggestion, we have supplemented the energy transfer properties of the ternary species **1/2/3** in the “good” solvent CH₂Cl₂, in which the monomeric state dominates. As shown in Figure R1a, when monomers **1–3** are dissolved in CH₂Cl₂, the energy transfer efficiencies (Φ_{ET}) for **1** and **2** are determined to be 37.1% and 16.7%, respectively. They are significantly lower than those in the “bad” solvent MCH, in which the supramolecular copolymeric state dominates (Φ_{ET} : 92.4% for **1** and 92.5% for **2**, Figure R1b]. Moreover, no fluorescence emission of **3** ($\lambda_{max} = 770$ nm as shown in Figure R1b) is observed for the ternary species **1/2/3** in CH₂Cl₂ due to the failure of cascade energy transfer. Hence, it is

evident that supramolecular copolymerization is necessary for the high energy transfer efficiency of the ternary species **1/2/3**.

Figure R1. Energy transfer experiments of the ternary species **1/2/3** (c : $8.0 \times 10^{-5} \text{ mol L}^{-1}$ for **1**, $1.6 \times 10^{-5} \text{ mol L}^{-1}$ for **2**, and $8.0 \times 10^{-6} \text{ mol L}^{-1}$ for **3**): a) in CH₂Cl₂, and b) in MCH.

2. *Reply to the second comment made by Reviewer 1 “In the last part: photo-responsiveness of each compound toward 460 nm and 525 nm are suggested be studied in more detail to give more information.”*

According to the reviewer’s comment, we have supplemented time-dependent absorption signal changes of supramolecular homopolymers derived from the individual compounds **1–3** upon exposure to 460 nm and 525 nm LED lamps. For **1**, the endoperoxidation photochemical reaction follows the first-order kinetics with a rate of 0.00265 s^{-1} under 460 nm light irradiation (Figures R2a,d). In comparison, they exhibit poor photo-oxidation capability when exposed to 525 nm light irradiation, as reflected by slight reduction of the anthracene’s absorption intensity (Figure R3a). In terms of **2**, the photo-oxygenation reaction follows to the second-order kinetics (Figure R3). When altering the irradiation wavelength from 460 nm to 525 nm, the photo-oxygenation rates increase from $0.0223 \text{ L mol}^{-1} \text{ s}^{-1}$ (Figures R2b,e) to $0.0506 \text{ L mol}^{-1} \text{ s}^{-1}$ (Figures R3b,d). For the individual compound **3**, it shows negligible self-sensitized photo-chemical reactions under both 460 nm (Figures R2c,f) and 525 nm (Figures R3c) light irradiation. Accordingly, the cascade energy transfer of the ternary supramolecular copolymers **1/2/3** can be modulated by the wavelength-selective photo-oxygenation reactions.

Figure R2. Absorption changes of supramolecular homopolymers derived from a) **1**, b) **2**, and c) **3** upon 460 nm light irradiation ($c: 5.0 \times 10^{-5} \text{ mol L}^{-1}$ in MCH), together with the kinetic rates of d) **1**, e) **2**, and f) **3**.

Figure R3. Absorption changes of supramolecular homopolymers derived from a) **1**, b) **2**, and c) **3** upon exposure to 525 nm light irradiation ($c: 5.0 \times 10^{-5} \text{ mol L}^{-1}$ in MCH), together with the kinetic rates of d) **2**, and e) **3**.

3. Reply to the third comment made by Reviewer 1 “Some grammar and spelling errors are found in the main text and supplementary information. For example, “dipole-dipole interactions” should be “dipole-dipole interactions”. The manuscript needs to be polished carefully.”

Thank the reviewer for the kind suggestion! We have carefully corrected these errors in the revised manuscript.

4. *Reply to the first comment made by Reviewer 2 “The authors describe beautiful results in a very interesting area of current science. They have used their detailed knowledge on supramolecular polymerization of Platinum-based monomers with the concept of energy transfer. Controlled energy transfer is of great importance for all kind of concepts. I like to strongly recommend publication, but a few points need to be improved before the manuscript can be accepted for publication. 1. The term Light Harvesting System is a bit misleading for this reviewer. It is for me a cascade energy transfer. Although it is with a very high yield – worth publishing in Nat. Comm.–but it does not harvest the light in some kind of reaction. Maybe, it is well accepted, but the general audience will probably expect more. The same type of nomenclature is used in the introduction. I would suggest the authors to make clearer what are the natural systems and what are mimics and where do they differ.”*

Thank the reviewer for the kind comment! We agree with the reviewer that the “light harvesting” is not an accurate concept in the current study, despite that many relating literatures in the energy transfer field use this term. Hence, we have removed the phrase “light harvesting” in the title. In addition, this term has been also replaced by “sequential energy transfer” or “energy transfer” in the main text and supplementary information.

5. *Reply to the second comment made by Reviewer 2 “The color coding in the figures is changing and makes following the manuscript difficult. See e.g. Figure 2a,b and Figure 3a,b.”*

According to the reviewer’s comment, we have kept the color coding consistent in the mentioned Figures.

6. *Reply to the third comment made by Reviewer 2 “The authors are using the model of Van der Schoot, many other are using today Mass Balance Models as developed by Markvoort and ten Eikelder (J. Phys. Chem. B. 2012, 116, 5291-5301).”*

According to the reviewer’s comment, we have employed the mass balance model to fit the supramolecular homopolymerization process of **1–3**, and acquired the thermodynamic parameters in the revised manuscript (see Table R1). The table has also been added in the main text (see Table 1 in the main text).

Table R1. Thermodynamic parameters of 1–3 obtained by temperature-dependent spectroscopic data^a

Monomer	ΔH_e (kJ mol ⁻¹)	ΔH_n (kJ mol ⁻¹)	ΔS (J mol ⁻¹ K ⁻¹)	ΔG_e^b (kJ mol ⁻¹)	σ^b
1	-74.5	-41.2	-160	-26.7	1.5×10^{-6}
2	-74.6	-44.1	-162	-26.3	4.6×10^{-6}
3	-95.0	-81.5	-222	-29.0	4.3×10^{-3}

^a The thermodynamic parameters were obtained by analyzing data acquired from CD spectroscopic experiments for **1–2** and absorption spectroscopic experiments for **3**. ^b Changes in Gibbs free energy of elongation (ΔG_e) and the cooperativity factor (σ) are reported for a temperature of 298 K.

7. *Reply to the fourth comment made by Reviewer 2 “Compound 3 is different from 1 and 2. Not only doesn’t it show CD, also the cooperativity is much less. It comes close to an isodesmic pathway.”*

Thank the reviewer for the kind comment! According to Table R1, the cooperativity factor of **3** is 4.29×10^{-3} , much larger than those of **1** and **2**. The data reflect the lower cooperativity for the supramolecular homopolymerization process of **3**. Moreover, the ΔG value for the supramolecular polymerization process of **3** at 298 K is determined to be -29.0 kJ mol⁻¹, which is larger than those of **1–2** (Table R1). As discussed in the main text, hydrogen bonding interactions play the prominent role for the **1–3** supramolecular homopolymerization processes. We rationalized that the additional non-covalent interactions may participate in the supramolecular polymerization process of **3**, because of the presence of the naphtho[2,3-*c*][1,2,5]selenadiazole unit in the inner core. DFT calculations show the larger dipole moment for the heteroacene monomer **3** than those of the acene monomers **1** and **2** (μ_D : 1.75 Debye of **3** versus 0.87 Debye for **1** and 1.20 Debye for **2**, Figure R4). It potentially forms dipole-dipole interactions between the neighboring molecules, thanks to the presence of intramolecular charge transfer interactions. When both hydrogen bonding and dipole-dipole interactions serve as the non-covalent driving forces in the supramolecular polymerization process of **3**, their subtle interplay may affect the non-covalent interaction mode and thereby lead to the lower cooperativity for supramolecular polymerization.

The conclusion is validated by DFT calculations. For homo-trimers **1**₃–**2**₃, the stacking directions of **1**₃ and **2**₃ exhibit slight deviation from the vertical direction (*z* axis). The

phenomena indicate low slipped angles between the stacked monomers (slipped angles: 9.9° for $\mathbf{1}_3$ and 6.5° for $\mathbf{2}_3$, Figure R4a,b). By contrast, $\mathbf{3}_3$ adopt slipped conformations (slipped angles for the three possible conformers of $\mathbf{3}_3$ are 40.7° , 53.5° , and 48.9° , Figure R4c). Hence, the contribution of dipole-dipole interactions may lead to the change of non-covalent stacking modes, which potentially result in the different cooperativity extent between $\mathbf{1}$ – $\mathbf{2}$ and $\mathbf{3}$ for the supramolecular homo-polymerization processes.

Figure R4. Graphic representation for the supramolecular polymerization modes and calculated dipole moments of a) $\mathbf{1}$, b) $\mathbf{2}$, and c) $\mathbf{3}$. All of the monomers and homo-trimers are optimized *via* DFT calculations. Depending on the spatial orientations of naphtho[2,3-*c*][1,2,5]selenadiazole units, $\mathbf{3}$ could possibly stack into three different conformers with minor differences in energies.

8. Reply to the fifth comment made by Reviewer 2 “Maybe the authors can say something about the thermodynamic parameters of the co-polymerization—especially when $\mathbf{3}$ is mixed with $\mathbf{2}$ and/or $\mathbf{1}$ and $\mathbf{2}$.”

According to the reviewer’s comment, we have supplemented supramolecular copolymerization behaviors of the binary species $\mathbf{2}/\mathbf{3}$ *via* CD spectroscopic measurements.

With the gradual addition of **3** into **2** (from 0 mol% to 10 mol%, by keeping the concentration of **2** at $8.0 \times 10^{-5} \text{ mol L}^{-1}$), the CD signals of **2** slightly decrease for their intensities (Figure R5a). Depending on the CD melting curves, T_e values slightly elevated with the increased ratios of **3** (from 315.2 K to 315.6 K, Figure R5b and Table R2). Meanwhile, the ΔH values increase from $-79.7 \text{ kJ mol}^{-1}$ to $-93.0 \text{ kJ mol}^{-1}$ (Table R2). It is consistent with the supramolecular homopolymerization processes, in which the ΔH value of **3** is higher than that of **2**. Nevertheless, the ΔG values are almost identical regardless of the amount of **3**, because of the increase of ΔS values with the higher loading of **3**. Overall, all of these phenomena exclude the self-sorting arrangement between **2** and **3**, and thereby support their supramolecular copolymerization tendency.

Figure R5. a) CD spectral changes of supramolecular copolymers **2/3** (c : $8.0 \times 10^{-5} \text{ mol L}^{-1}$ for **2**) in MCH upon varying the ratio of **3** from 0 mol% to 10 mol%. b) CD melting curves and nonlinear fitting for melting curves of **2/3** via the mass balance model.

Table R2. Thermodynamic parameters of supramolecular copolymers **2/3**

Molar ratio of 3	T_e (K)	ΔH (kJ mol^{-1})	ΔS ($\text{J mol}^{-1} \text{ K}^{-1}$)	NP (kJ mol^{-1})	ΔG at 298 K (kJ mol^{-1})	σ (298 K)
0 mol%	315.2	-79.7	-174	-31.8	-27.7	2.70×10^{-6}
5 mol%	315.7	-83.5	-186	-30.9	-28.1	3.89×10^{-6}
10 mol%	315.6	-93.0	-216	-30.7	-28.6	4.16×10^{-6}

9. Reply to the first comment made by Reviewer 3 “In the current paper, the authors reported a novel artificial supramolecular SET - LHSs inspired, constructed via the supramolecular copolymerization of σ -platinated acenes. By comparing the experimental measurement with DFT and TDDFT calculation results, the authors showed

that through minor structural variations of acenes, the transitions of vibronic absorption and emission bands could be readily tuned from the visible to NIR region. This result sounds reliable. However, the reviewer pays attention that in the “Result, Spectroscopy of 1–3 in monomeric state” section, the maximum peak in both of the Absorption spectra (Fig 2a) and electronic transition spectrum (Supplementary Fig 1~3) are not located at the HOMO-LUMO transition region. These data show in the plot but lack discussion in the main text. I would suggest the authors provide some discussions about those peaks. Other than that, the paper is in good shape and suitable for publishing in Nature Communications.”

Thank the reviewer for the kind comment! Compounds **1–3** display two absorption bands in the UV region: namely a low-energy broad band at 308~395 nm, together with a high-energy strong and sharp band. The detailed discussion of the spectroscopic properties is listed as follows.

The former bands range from 308 nm to 395 nm [λ_{\max} : 332 nm ($\epsilon = 7.93 \times 10^4 \text{ M}^{-1} \text{ cm}^{-1}$) for **1**; 334 nm ($\epsilon = 6.76 \times 10^4 \text{ M}^{-1} \text{ cm}^{-1}$) for **2**; 328 nm ($\epsilon = 9.21 \times 10^4 \text{ M}^{-1} \text{ cm}^{-1}$) for **3**]. With reference to the previous literatures of *trans*-[Pt(PEt₃)₂(C≡CR)₂] complexes (e.g. *Chem. Commun.*, 2013, 49, 6977), they mainly arise from the intra-ligand (IL) [$\pi \rightarrow \pi^*(\text{C}\equiv\text{CR})$] transitions, with minor contributions from the metal-to-ligand charge transfer (MLCT) [$d(\text{Pt}) \rightarrow \pi^*(\text{C}\equiv\text{CR})$] transitions. TD-DFT calculations (Figure R6) confirm that the IL transitions of amide ligands mainly contribute to the compositions. In addition, **3** exhibits an additional absorption at 386 nm ($\epsilon = 6.93 \times 10^4 \text{ M}^{-1} \text{ cm}^{-1}$). Based on the DFT calculation, it may be assigned to the high-energy transition of naphtho[2,3-*c*][1,2,5]selenadiazole unit perturbed by Pt(II) ions (Figure R7a).

In terms of the latter bands, the maxima absorption signals locate at 282 nm ($\epsilon = 1.11 \times 10^5 \text{ M}^{-1} \text{ cm}^{-1}$) for **1**, 303 nm ($\epsilon = 1.18 \times 10^5 \text{ M}^{-1} \text{ cm}^{-1}$) for **2**, and 295 nm ($\epsilon = 1.01 \times 10^5 \text{ M}^{-1} \text{ cm}^{-1}$) for **3**. It mainly originates from high-energy transitions of (hetero)acene units according to the previous literatures (*Nat. Commun.* **2018**, 9, 3977; *Organometallics* **2013**, 32, 1620; *Org. Lett.*, **2013**, 15, 666).

Figure R6. Electron density distributions for HOMO- m ($1 \leq m \leq 2$) and LUMO+ n ($1 \leq n \leq 2$) of a) **1**, b) **2**, and c) **3**.

Figure R7. Electron density distributions for a) HOMO and LUMO+3 orbitals of **3**, b) HOMO and LUMO- n ($7 \leq n \leq 8$) orbitals of **1**, and c) HOMO and LUMO- n ($11 \leq n \leq 12$) orbitals of **2**.

10. *Reply to the second comment made by Reviewer 3 “... What the excited state (S_2 , S_3 ?) and transition ($HOMO-1 \rightarrow LUMO+1$) are related to these peaks? Are those peaks also captured in the experimental signals? ... I would suggest the authors provide some discussions about those peaks.”*

According to the reviewer’s comment, we have employed TD-DFT calculations to understand the origin of UV-region transitions. For the simulated spectrum of **1**, the transition energy of excited state 2 ($\lambda = 357.73$ nm, $f = 1.03$) is close to experimental result (λ

= 332 nm). TD-DFT calculations support that excited state 2 originates from HOMO- m →LUMO+ n ($1 \leq m \leq 2$, $1 \leq n \leq 2$) transitions. For HOMO- m orbitals ($1 \leq m \leq 2$), electronic densities are mainly delocalized on the 4-alkynylaniline units of the amide ligands (Figure R6a), with minor contributions from Pt(II) ions. Upon photo-excitation, the electron densities partially migrate to benzoyl units of the amide ligands in LUMO+ n orbitals (Figure R6a), accompanied by the decreased electronic transitions of Pt(II) ions. Therefore, these transitions primarily arise from IL transitions of the amide ligands, with some admixture of MLCT transitions. Likewise, HOMO- m →LUMO+ n ($1 \leq m \leq 2$, $1 \leq n \leq 2$) transitions are also important compositions for **2** (excited state 2: $\lambda = 365.88$ nm, $f = 1.1399$; excited state 3: $\lambda = 356.70$ nm, $f = 0.3342$) and **3** (excited state 2: $\lambda = 362.73$ nm, $f = 0.2046$; excited state 3: $\lambda = 359.39$ nm, $f = 0.3979$) (Figure R6b,c). Therefore, TD-DFT calculations support that **1–3** possess similar transition mechanisms for the absorptions at *ca.* 360 nm.

Notably, the simulated spectrum of **3** shows that another excited state ($\lambda = 388.68$ nm, $f = 1.2799$) could be generated within the range of 350–400 nm. The calculation result is consistent with the experimental spectroscopic data (λ_{max} : 386 nm). According to the TD-DFT calculations, the band primarily arises from HOMO→LUMO+3 transition (contribution: 89.4%). Since the electron densities of HOMO and LUMO+3 orbitals are delocalized on alkynylated naphtho[2,3-*c*][1,2,5]selenadiazole and Pt(II) ions (Figure R7a), the band is assigned to high-energy transition of naphtho[2,3-*c*][1,2,5]selenadiazole unit perturbed by Pt(II) ions according to TD-DFT results.

For the shorter wavelengths, the transition energies of **1–2** located at around 313 nm [Note: since **3** shows rather weak transitions ($f < 0.05$), we would not discuss transitions of **3** thereafter]. For **1**, the HOMO→LUMO+ m ($7 \leq m \leq 8$) transitions are the primary compositions (contribution: 79.7%) for excited state 4 ($\lambda = 313.87$ nm, $f = 0.078$). Upon photo-excitation, the electron densities migrate from the alkynylated anthracene [with minor contribution from Pt(II) ions] to the amide ligands and Pt(PEt₃)₂ moieties (Figure R7b). Therefore, ligand-to-ligand charge transfer (LLCT) and ligand-to-metal charge transfer (LMCT) transitions contribute to the excited state 4. In terms of **2**, LLCT/LMCT transitions from alkynylated tetracene to amide ligands and Pt(PEt₃)₂ units are also observed for HOMO→LUMO+ m ($11 \leq m \leq 12$) transitions ($\lambda = 313.18$ nm for the excited state 4, $f = 0.0672$, Figure R7c).

Next, we turned to the highest transition regions in 290~300 nm (λ : 282 nm for **1** and 303 nm for **2**). For **1**, the HOMO- m →LUMO+ n ($1 \leq m \leq 2$, $3 \leq n \leq 4$) transitions are mainly responsible for the formation of excited state 5 ($\lambda = 292.52$ nm, $f = 0.4860$). For LUMO+ n ($3 \leq n \leq 4$) orbitals, the electronic densities are primarily delocalized on Pt(II) ions and benzoyl units of amide ligands (Figure R8a). In particular, electron densities on the benzoyl moieties and Pt(II) ions are significantly higher than those of HOMO- m ($1 \leq m \leq 2$), whereas it is decreased for 4-alkynylaniline units of the amide ligands (Figure R8a). Hence, electronic transitions with LMCT and IL characters contribute to the formation of excited state 5. For the structurally similar compound **2**, TD-DFT calculations also support that the electronic transitions with LMCT and IL characters [HOMO- 1 →LUMO+ m ($4 \leq m \leq 6$) transitions] contribute to the generation of excited state 5 ($\lambda = 296.06$ nm, $f = 0.0825$, Figure R8b).

In summary, TD-DFT calculations are reliable methods to predict the absorption transitions of **1–3**. Some inconsistency exists for the higher energy UV-region transitions between the theoretical and experimental results. For the lower energy UV-region transitions, TD-DFT calculations unambiguously support the presence of IL/MLCT transitions of amide ligands, which are in accordance with the spectroscopic results.

Figure R8. Electron density distributions for a) HOMO- m ($1 \leq m \leq 2$) and LUMO+ n ($3 \leq n \leq 4$) orbitals of **1**, b) HOMO and LUMO- n ($4 \leq n \leq 6$) orbitals of **2**.

11. Reply to the third comment made by Reviewer 3 “...Although these peaks appear at higher energy levels, they are still within the high Violet to low UV wavelengths region and may affect the reported spectral red-shifts phenomenon.... I would suggest the authors provide some discussions about those peaks.”

Thank the reviewer for the kind comment! Generally, for the donor- π -acceptor organic molecules, electron communication exists between the donor (with high transition energy) and acceptor (with low transition energies) units *via* π -conjugated bridges, leading to the red-shifting of absorption bands upon excitation of the donor unit. However, the transition mechanisms of platinum(II) acetylide molecules reported in the current manuscript are different with those of the traditional donor- π -acceptor molecules. When donor and acceptor units are linked *via* the platinum(II) ions, the *5d* orbitals of Pt(II) and *p*-orbitals of the alkynylated ligands poorly overlap with each other. As a result, platinum(II) ions fail to act as the π -conjugated bridges. It restricts electronic communication and energy transfer between donor/acceptor units, as widely reported in the previous literatures (*e.g.*, *Organometallics*, **2012**, *31*, 7522; *Organometallics*, **2013**, *32*, 7283). Hence, in the current systems the peripheral amide ligands do not interfere with HOMO \rightarrow LUMO transitions of the inner (hetero)acene units. Additionally, the high-energy transition bands of (hetero)acene units locate at UV region. Excitation of these bands results in the formation of high energy singlet excited states (S_n states), which undergo relaxation processes ($S_n\rightarrow S_1$) to the first singlet excited states (S_1 states). Hence, it exerts no impacts on the $S_0\rightarrow S_1$ (corresponding to HOMO \rightarrow LUMO) transitions of the (hetero)acene units.

With reference to the previous literatures (*Organometallics*, **2011**, *30*, 6383; *Organometallics*, **2013**, *32*, 1620; *Organometallics*, **2010**, *29*, 2422), the spectral red-shifting of the HOMO \rightarrow LUMO transitions can be interpreted by the metal- π interactions. For the TIPS (triisopropylsilyl) substituted precursors **7–9** (Figure R9a), the vibronic absorption bands red-shift from anthracene *via* tetracene to naphtho[2,3-*c*][1,2,5]selenadiazole derivatives. When TIPS substituents are replaced by [Pt(PEt₃)₂X]⁺ moieties (X = alkynylated amide ligand, Figure R9a), HOMOs/LUMOs of the alkynylated (hetero)acenes are perturbed by the molecular orbitals of [Pt(PEt₃)₂X]⁺ moieties *via* metal- π interactions, giving rise to destabilization of the HOMOs/LUMOs of TIPS substituted precursors (Figure R9b). Since the MOs of [Pt(PEt₃)₂X]⁺ have better energy matching with the HOMOs of (hetero)acenes, they are destabilized to a large extent than the LUMOs, as revealed by the larger changes of HOMOs upon platination of (hetero)acene units (Figure

R9b). In particular, the destabilization effect endows HOMOs of **1–3** with the similar energy levels (E_{HOMO} : -4.129 eV for **1**, -4.042 eV for **2**, and -4.298 eV for **3**), while large fluctuations exist in the energy levels of LUMOs [E_{LUMO} : -1.342 eV for **1**, -1.772 eV for **2**, and -2.542 eV for **3**, Figure R9b]. Therefore, the HOMO–LUMO gaps reduce from **1** (2.787 eV, 444.9 nm) *via* **2** (2.270 eV, 546.3 nm) to **3** (1.756 eV, 706.2 nm) (Figure R9b). Therefore, it can be concluded that the metal- π interactions play vital roles for the red-shifting of absorption bands in **1–3**.

Figure R9. a) Structures for TIPS and [Pt(PEt₃)₂X] substituted compounds bearing anthracene, tetracene, and naphtho[2,3-*c*][1,2,5]selenadiazole units. b) Energy level diagram for TIPS substituted precursors **7–9** [optimized at B3LYP/6-31G(d) level] and monomers **1–3**. The energy levels of HOMOs and LUMOs are marked by red and blue lines.

Some of the data have been added in the main text and supplementary information (see the revised manuscript with highlights). We are hopeful that it now meets your standards for acceptance. Thanks a lot!

Feng Wang

REVIEWERS' COMMENTS

Reviewer #1 (Remarks to the Author):

This is the revised version of previous submission. The authors have answered all the questions which the reviewers were concerned. Therefore this referee recommend it to be published on the nature Communication after the minor reversion or revise them during the Proof-reading :

This referee checked the supporting information, and realize that the ^1H NMR of compounds 2 and 3 need to be revised. In general, For ^1H NMR of molecules, the J-Coupling constants for different multiplet of Signals (Singlet, Doublet, Triplet, Quartet et al) should be given. For the Chemical Shift of Signals (s,d, t, q) is a certain spot, not a range. please revise the small bug.

Reviewer #2 (Remarks to the Author):

I am highly impressed by the thorough revisions made and I like to congratulate the authors with an outstanding paper

Thank you and all of the reviewers for their insightful comments on this paper (NCOMMS-21-45163)! We have made the following corrections in response to the reviewers' comments.

*1. Reply to the first comment made by Reviewer 1 "This is the revised version of previous submission. The authors have answered all the questions which the reviewers were concerned. Therefore this referee recommend it to be published on the nature Communication after the minor reversions or revise them during the Proof-reading: This referee checked the supporting information, and realize that the ^1H NMR of compounds **2** and **3** need to be revised. In general, For ^1H NMR of molecules, the J-Coupling constants for different multiplet of signals (Singlet, Doublet, Triplet, Quartet et al) should be given. For the Chemical Shift of Signals (s,d, t, q) is a certain spot, not a range. please revise the small bug."*

Thank the reviewer 1 for the kind comment on this manuscript. We have added the coupling constants for aromatic hydrogen atoms of **2** and **3** in the supplementary file. The incorrect writing patterns for chemical shifts of singlet, doublet, triplet, quartet resonance signals have been revised as well.